# Cardiac PET Imaging of ATP Binding Cassette (ABC) Transporters: Opportunities and Challenges

**DOI:** 10.3390/ph16121715

**Published:** 2023-12-11

**Authors:** Wanling Liu, Pascalle Mossel, Verena Schwach, Riemer H. J. A. Slart, Gert Luurtsema

**Affiliations:** 1Department of Nuclear Medicine and Molecular Imaging, University Medical Center Groningen, University of Groningen, Hanzeplein 1, 9713 GZ Groningen, The Netherlands; w.liu01@umcg.nl (W.L.); p.mossel@umcg.nl (P.M.); 2Department of Applied Stem Cell Technologies, TechMed Centre, University of Twente, 7500 AE Enschede, The Netherlands; v.schwach@utwente.nl; 3Department of Biomedical Photonic Imaging, University of Twente, 7500 AE Enschede, The Netherlands

**Keywords:** ABC transporters, positron emission tomography, heart, P-glycoprotein, pharmaceuticals

## Abstract

Adenosine triphosphate binding cassette (ABC) transporters are a broad family of membrane protein complexes that use energy to transport molecules across cells and/or intracellular organelle lipid membranes. Many drugs used to treat cardiac diseases have an affinity for these transporters. Among others, P-glycoprotein (P-gp) plays an essential role in regulating drug concentrations that reach cardiac tissue and therefore contribute to cardiotoxicity. As a molecular imaging modality, positron emission tomography (PET) has emerged as a viable technique to investigate the function of P-gp in organs and tissues. Using PET imaging to evaluate cardiac P-gp function provides new insights for drug development and improves the precise use of medications. Nevertheless, information in this field is limited. In this review, we aim to examine the current applications of ABC transporter PET imaging and its tracers in the heart, with a specific emphasis on P-gp. Furthermore, the opportunities and challenges in this novel field will be discussed.

## 1. Introduction

Adenosine triphosphate binding cassette (ABC) transporters are a diverse group of membrane protein complexes that use ATP hydrolysis energy to transport chemicals across cells and other biological membranes [1]. In humans, 49 ABC transporters have been discovered. They transport a variety of compounds including proteins, lipids, and drugs [2,3], and serve an important function in protecting organs such as the intestines, liver, brain, eyes, testicles, and heart against hazardous chemicals [4].

Recently, interest in ABC transporters and cardiology has been increasing [5,6,7]. Researchers have discovered their expression in the heart, including ABCB1 (P-glycoprotein, P-gp, multidrug-resistant transporter, MDR1) [8,9], ABCC5 (multidrug resistance proteins 5, MRP5) [10], and ABCG2 (breast cancer resistance protein, BCRP) [11]. They act as a functional barrier between the blood and other cells within the heart, such as cardiomyocytes and myofibroblasts, limiting medication and the entry of foreign substances into the heart.

ABCB1(P-gp) is one of the most essential efflux transporters [12]. Many commonly employed cardiovascular pharmaceuticals, such as β-blockers, cardiac glycosides, and Ca^2+^ channel blockers, are transported by P-gp and are known as P-gp substrates. P-gp transports these substrates, and any changes in P-gp function can impact the bioavailability of these pharmaceuticals. When P-gp function is reduced, there is a potential for (toxic) drug accumulation within the heart tissue. This increased concentration of drugs in the cardiac tissue may cause a decline in cardiac function, a phenomenon referred to as cardiotoxicity. For example, doxorubicin is a P-gp substrate that has been shown to accumulate in the hearts of mice lacking P-gp and cause dose-dependent heart failure in cancer patients [13,14]. When P-gp function is induced or activated, less of a drug may be available at the target organ, making the drug less effective. Therefore, the evaluation of P-gp function through imaging techniques may be crucial for the avoidance of cardiotoxicity in drug interactions and might play an important role in personalized medicine [15,16].

Assessment of cardiac ABC transporters in vivo may benefit the study of drug availability in therapy, and guide and predict effective treatment in clinical practice. Currently, the expression of ABC transporters in the heart tissue is studied primarily in vitro, using reverse transcription polymerase chain reaction (RT-PCR), immunohistochemistry, and Western blot [17]. Additionally, some studies have demonstrated alterations in the function and variations in expression of ABC transporters in the heart. The function of ABC transporters can be affected by inhibitors or inducers, or in some pathological situations, including dilated and ischemic cardiomyopathy [18,19]. It is evident that in vivo measurements using molecular imaging of the living heart can be beneficial since they provide real-time, non-invasive quantitative and functional information.

Positron emission tomography (PET) is a powerful imaging technique that uses components labeled with radionuclides, known as radiopharmaceuticals or radiotracers, to scan and measure biochemical processes in vivo [20]. The technique makes it possible to image and quantify, in vivo, sensitively, and non-invasively biochemical and (patho)physiological processes. Additionally, functional P-gp at the blood–brain barrier (BBB) has been effectively studied using PET tracers such as [^11^C]verapamil, [^18^F]MC225, and [^11^C]metoclopramide [21,22]. Therefore, employing PET imaging to assess cardiac P-gp will offer valuable insights for drug development and enhance the precision of medication usage. However, the current understanding of this area is limited. This review aims to explore the current applications of ABC transporter PET imaging, specifically focusing on P-gp in the heart. Additionally, we will discuss the opportunities and challenges in this evolving field.

## 2. ABC Transporters of the Heart

### 2.1. ABC Transporters

ABC transporters are membrane proteins that facilitate a variety of ATP-driven transport activities and typically consist of a pair of nucleotide-binding domains (NBDs) and two transmembrane domains (TMDs). The TMDs are responsible for substrate specificity, while the NBDs exhibit a high degree of conservation across various ABC transporters. Positioned in the cytoplasm, the NBDs play a key role in transferring energy to facilitate the transport of the substrate across the membrane [23,24]. In 1976, Juliano et al. [25], through surface labeling studies, found a 170 K Daltons cell surface glycoprotein in the plasma membrane of ovary cells in colchicine-resistant Chinese hamsters. They designated it the P-glycoprotein (P-gp), which is the first ABC transporter. There are currently 49 human ABC transporters known and have been classified into seven subfamilies (ABCA-ABCG), according to sequence homology, domain structure, and functional similarity [24]. In addition to simple sugars, fatty acids, nucleosides, and amino acids, they also assist in the active transport of complex polysaccharides, lipids, oligonucleotides, and proteins [24]. ABC transporters demonstrate expression in various tissues and organs such as the liver, intestine, kidney, heart, and brain. They play crucial roles in drug absorption, distribution, and excretion, and are associated with many human diseases, such as deficiencies in sterol transport deficiencies, and mitochondrial iron homeostasis [23].

### 2.2. ABC Transporter Families in Cardiac Tissue

Many studies have demonstrated how ABC transporters are expressed in the heart [11,17,18,26] and how they facilitate the transport of additional substrates from the heart to plasma. Table 1 summarizes the expression and functionality of these transporters, as well as diseases caused by changes in the transporters.

In brief, ABCB1(P-gp), ABCC1(MRP1), and ABCG2 (BCRP) serve as defense mechanisms against anthracyclines, mitoxantrone, and other cardiotoxic drugs in the heart [11,18,27].

ABCA1, 2, 5, 8–10, and ABCG1 promote the excretion of excess intracellular cholesterol and phospholipids, and the production of high-density lipoprotein (HDL) [7,28,29,30,31,32,33,34,35,36]. Disorders in lipoprotein metabolism can contribute to the deposition of fatty substances in medium and large arteries, thereby causing atherosclerosis and potential vascular obstruction resulting in brain or cardiac infarction [37]. Coronary heart disease (CHD) negatively influences heart function when blood flow in the coronary arteries is reduced or blocked as a result of obstructive atherosclerosis [38].

ABCC5 serves as an efflux pump for cyclic nucleotides, particularly 3′,5′-cyclic guanosine monophosphate (cGMP). This molecule holds significant importance as one of the primary second messengers pivotal in the regulation of cardiac contractility and electrophysiology [26]. ABCC5 is found in cardiomyocytes as well as endothelial cells, and its expression is enhanced in ischemic cardiomyopathy, which is defined as systolic left ventricular failure in the presence of obstructive coronary artery disease. As a result, ABCC5-mediated cellular export might be a unique, disease-dependent mechanism for cGMP removal from cardiac cells.

ABCC8 and ABCC9 are vital components of the ATP-sensitive potassium (K_ATP_) channels in the heart [39,40,41]. These K_ATP_ channels serve as unique cellular energy sensors, safeguarding cardiomyocytes, particularly during conditions such as ischemia [42].

**Table 1 pharmaceuticals-16-01715-t001:** Overview of ABC transporters expressed in heart tissue.

	Expression	Function	Disease	Ref.
ABCA1	Macrophages in (coronary) arteries	Mediate nascent HDL biogenesis and allow for the excretion of excess intracellular cholesterol and phospholipids	Dyslipidemia; Atherosclerosis; Coronary heart disease	[7,28,29]
ABCA2	Heart; monocytes and macrophages	A sterol transporter maintains homeostatic levels of cholesterol.	Atherosclerosis; Alzheimer’s disease	[35,36]
ABCA5	Macrophages in (coronary) arteries	A macrophage cholesterol efflux.	Atherosclerosis; Coronary heart disease	[31,43]
	Cardiomyocytes	Involved in the processing of autolysosome.	Lysosomal diseases in heart; Dilated cardiomyopathy	[44]
ABCA8	Heart; macrophages	Cholesterol efflux and HDL transshipment.	Atherosclerosis; Coronary heart disease	[32,33,34]
ABCA9	Heart; macrophages	Works in macrophage lipid homeostasis and monocyte differentiation.		[45]
ABCA10	Heart; macrophages	Serve functions in lipid trafficking.		[46]
ABCB1 (P-gp, MDR1)	Endothelial cells in capillaries and arterioles	Acts as an effective barrier between the blood and the cardiac myocytes. Shields heart tissue from the cardiac toxicity of some medicines.	Dilated cardiomyopathy	[8,9,18]
ABCB10	Heart; inner mitochondrial membrane	Affects heme synthesis and mitochondrial iron transportation.	Myocardial ischemia/reperfusion injury	[47,48]
ABCC1 (MRP1)	Heart; vascular endothelial cells	Facilitates the transport of conjugates and hydrophobic substances; controls vascular endothelial homeostasis and blood pressure.	Anthracycline-induced cardiotoxicity; Hypertension; Atherogenesis	[27,49]
ABCC5 (MRP5)	Cardiomyocytes; vascular	An organic anion export pump removes cGMP from cardiac cells.	Ischemic and dilated cardiomyopathy	[18,26]
ABCC8 (SUR1)	Cardiac atrial K_ATP_ channels	Functional role in sarcolemma ATP-sensitive potassium channels.	Myocardial ischemia/reperfusion injury	[41,50]
ABCC9 (SUR2)	Cardiac ventricular myocytes	Essential element of the sarcolemma K_ATP_ channel in cardiac ventricular myocytes.	Dilated cardiomyopathy; Cantú syndrome	[39,40]
ABCG1	Macrophage; epicardial and subcutaneous adipose tissue	Mediates cholesterol transport to the HDL fraction.	Atherosclerosis; Coronary heart disease	[29,51]
ABCG2 (BCRP)	Arterioles; endothelial cells side population (SP) stem cells	Influences the survival, migration, and tube formation of microvascular endothelial cells in the heart.	Myocardial ischemia/reperfusion injury dilative; Ischemic cardiomyopathies	[11,19,52]

### 2.3. P-gp Expression in Cardiac Tissue

Among all ABC transporters, P-gp is found to be the most relevant to cardiovascular therapy [12]. P-gp, a 170 kDa plasma membrane protein, is a well-characterized efflux pump presenting at the arterioles and capillaries of the heart [18,53]. In human heart vessel cells, the function of P-gp is similar to that of capillary endothelial cells at the BBB [18]. It functions as a barrier between the blood and cardiomyocytes, thus limiting the bioavailability and absorption of P-gp substrate drugs, such as digoxin, into the heart tissue. It is therefore a key regulator of therapeutic effects, and its expression in the human heart may have significant therapeutic implications. Similarly, P-gp also regulates drugs that have cardiotoxic effects, such as doxorubicin, which might lead to dose-dependent heart failure, arrhythmias, or myopathies in patients [54]. In the case of another P-gp substrate, vinblastine, which is a chemotherapeutic agent with cardiotoxicity, the drug concentration in the heart was 10-fold higher in P-gp knockout mice than in wild-type mice 24 h after injection of the tracer [55]. Cardiac P-gp thus shields cardiac tissue against the toxicity produced by some medicines.

## 3. Regulation of P-gp in the Heart

The up and down-regulation of P-gp activity within the body can significantly impact drug bioavailability, renal clearance, and peripheral tissue distribution. Therefore, it is crucial to understand how P-gp undergoes changes and is regulated by different influencing factors. The pivotal role of P-gp in drug transport across bio-membranes is being increasingly acknowledged. Compounds that are transported are referred to as substrates, whereas chemicals that impact the function of the transporter are referred to as inhibitors and inducers. In the heart, P-gp facilitates the transport of many cardiovascular drugs that act as substrates, thereby regulating their localized concentration within the heart. The P-gp function can be affected by a diverse range of factors such as disease, age, gender, circadian rhythm, and dietary choices [56,57,58], as well as drugs that act as inducers or inhibitors of P-gp, leading to changes in bioavailability of these cardiovascular drugs. The following sections will provide further insight into the substrates of P-gp, and the diverse set of factors involved in regulating its function.

### 3.1. Substrates of P-gp

There are no evident structural similarities between P-gp substrates [23]. According to the FDA drug interaction guidance [59], the P-gp substrates test must be performed on all investigational medicines. Table 2 lists several common medications that have been identified as P-gp substrates. Digoxin is used as a cardiac glycoside to treat heart failure and cardiac arrhythmias. Due to its restricted therapeutic window, digoxin may lead to serious adverse drug reactions ranging from arrhythmia recurrence, failure of device-based treatment, cardiac failure, and death. [60]. In a clinical trial [60], digoxin had a reduced renal clearance and an increased half-life in healthy human volunteers after two weeks of pretreatment with the P-gp inhibitors ritonavir and saquinavir. Electrocardiographic measurements also showed a trend for a longer PR interval in this trial [60]. In a recent study, Mualla et al. [61] found that inhibiting cardiac P-gp increased digoxin accumulation in the heart. Notably, they directly examined the effects of P-gp on the heart using the Langendorff perfusion technique, enabling pharmacokinetic and pharmacodynamic research at the organ level and identifying the function of P-gp in drug bioavailability inside the cardiac tissue.

### 3.2. Inhibitors of P-gp

The presence of inhibitors, inducers, and/or activators, together referred to as modulators, affects P-gp expression and function levels. P-gp inhibitors are chemicals that reduce the transport of substrates by P-gp. They impede P-gp function by interfering with ATP hydrolysis, changing P-gp expression, or competing for binding sites in a reversible or irreversible manner.

Extensive research and development efforts have been dedicated to inhibitors in the field of oncology. This is because of the association of P-gp with multidrug resistance (MDR), a form of acquired resistance observed in cancer cells in response to chemotherapeutic treatments [66]. Based on their selectivity, affinity, and toxicity, P-gp inhibitors are divided into four generations. The first generation includes verapamil, quinidine, and cyclosporin A. The second generation includes dexverapamil and dexniguldipine, which are more selective and have fewer adverse effects. Third-generation molecules, like Tariquidar, exhibit a strong affinity for P-gp at nanomolar doses. Fourth-generation compounds, including flavonoids, alkaloids, and terpenoids, have been produced to perform better in terms of efficacy and toxicity [67].

In clinical practice, first-generation P-gp inhibitors play a crucial role in drug–drug interactions (DDI). This implies that the effects of those drugs can modify the impact of others, thereby influencing their respective therapeutic domains. Additionally, these treatments may be long term. For example, verapamil is known as a calcium channel blocker and is used in cardiovascular diseases, including angina pectoris and hypertension. As verapamil also acts as a substrate and as an inhibitor, depending on the dose, it reduces P-gp function in organs and tissues during treatment. For example, clinical research found that administering omadacycline after a single oral dosage of 240 mg verapamil ER increased systemic exposure to omadacycline by 14–25% as compared to omadacycline alone [68]. If a combination drug is a P-gp substrate, its concentration in tissue may increase, leading to potential toxicity. For instance, the inhibitor nilotinib can effectively restrain the P-gp-mediated efflux of anticancer drugs. This inhibition leads to an increased concentration of anticancer drugs within the tumor, thereby enhancing the sensitivity of anticancer medication; however, this inhibits cardiac P-gp function and at the same time extends the presence of both doxorubicin and doxorubicinol in cardiac tissue, thereby increasing the risk of cardiotoxicity [69].

The relationship between the substrate and inhibitor is complex and worthy of note. Several drugs, such as quinidine, verapamil, warfarin, and atorvastatin, are also known as inhibitors. The effect of these inhibitors is dose dependent. For example, tariquidar in lower concentrations inhibits P-gp while acting as a BCRP substrate. However, when its concentration is higher than 100 nM it inhibits both P-gp and BCRP.

### 3.3. Inducers and Activators of P-gp

P-gp inducers enhance both its expression and function. The inducers are facilitated through the pregnane xenobiotic receptor (PXR) and the constitutive androstane receptor (CAR). Both PXR and CAR are nuclear receptors. Following activation by P-gp inducers, they bind to the transcriptional binding sites of P-gp, thereby promoting increased expression [70]. Activators, as distinguished from inducers, enhance P-gp function quickly without affecting protein expression levels. In an in vitro investigation, Vilas-Boas et al. [71] discovered that rifampicin, known as a PXR agonist, dramatically enhanced P-gp expression after 72 h, while its derivatives RedRif increased both P-gp expression and function. Chemicals like RedRif may be effective as an antidote to P-gp cytotoxic substrates. Furthermore, in an animal study [72], Dexamethasone raised Mdr1b gene expression in the heart by 1.4 fold. However, most P-gp inducers typically decrease drug bioavailability in the body, which may impact therapeutic efficacy [70].

### 3.4. Heart Diseases Affecting P-gp Function

Some studies have shown that diseases can alter P-gp expression and function. Konrad et al. [18] evaluated the mRNA and protein levels of human P-gp using 15 excised left ventricles obtained from orthotopic heart transplantation procedures. Lower expression levels of P-gp are exhibited in nonischemic dilated cardiomyopathy, characterized by impaired heart systolic function without interruption of perfusion [73,74]. This differs from ischemic cardiomyopathy, where systolic left ventricular dysfunction results from the interruption of blood perfusion [75]. This result could not be explained by differences in drug therapy. In a recent study by Auzmendi et al. [8], it was discovered that seizures induce ischemia–reperfusion injury, and this phenomenon is associated with an up-regulation of P-gp in cardiomyocytes. The observed increase in P-gp expression can be ascribed to the activation of hypoxia-inducible factor 1α (HIF-1α). Serving as a crucial transcriptional regulator, HIF-1α plays a central role in cellular adaptation to hypoxic conditions by overseeing the transcriptional activation of various genes, including ABCA1.

As for other ABC transporters, their expression is also influenced by disease. The levels of ABCG2 mRNA in cardiomyopathic hearts (both dilative and ischemic cardiomyopathy) were higher than in non-failing hearts [19]. MRP5 expression was found to be increased in ischemic cardiomyopathy [26].

### 3.5. Other Factors Affecting P-gp Function

Apart from drugs and diseases, several factors may contribute to the significant interindividual variability of P-gp, including gender, age, genes, diet, and circadian rhythm. Baris et al. [76] investigated the effect of carvedilol on blood digoxin levels and its gender differences in heart failure patients. They observed that men show higher P-gp function than women. Some researchers discovered a decline in brain P-gp function with aging [77]. Burk et al. [78] describe the discovery and distribution of 15 polymorphisms in the human ABCB1 gene that codes for P-gp, one of which, exon 26 (C3435T), is associated with P-gp expression levels and function in vivo. The C3435T mutation frequency is highly impacted by ethnicity, with African groups having higher frequencies than Caucasian and Asian populations [79]. Circadian rhythm [56,57] and dietary components [58] such as curcumin, garlic, and green tea also contribute to P-gp modulation. Moreover, continuous and repeated administration of P-gp substrate drugs also led to an increase in P-gp [80]. As such, accurate and individualized evaluation of P-gp expression and function in vivo is important for future research.

## 4. PET Imaging

PET stands as a valuable imaging technique for the in vivo assessment and quantification of P-gp function in both animals and humans. Kinetic modeling is used to derive in vivo measures from radiotracer concentration–time curves in plasma and tissue. Micro-parameters like K1, k2 and the volume of distribution (V_T_), which reflects the P-gp function, can be calculated. PET is a non-invasive imaging technique and finds extensive application in both preclinical and clinical research, making it a valuable tool for translational studies focused on P-gp substrates and regulatory drugs. PET proves to be highly beneficial and is well-validated for non-invasive clinical investigations of vital organs such as the heart [16,81].

### 4.1. Methods to Measure ABC Transporters Function with PET

Quantification of PET data can be achieved through several mathematical methods, with the standardized uptake value (SUV) emerging as the predominant measure for analysis. SUV is determined by dividing the ratio of radiotracer concentration within a specific region of interest (ROI) or a collection of voxels by the administered radioactive dose. This metric offers advantages such as obviating the need for arterial cannulation and shorter scan durations. In addition, compartmental modeling (CM), also known as kinetic modeling, is regarded as the “gold standard” in PET quantification. In CM, it is thought that the PET radiotracer is transferred across compartments. The kinetic parameters or micro-parameters can be calculated using a numerical process; these parameters can then be utilized to produce physiological measurements of interest [82].

To quantify transporter function in vivo, CM is the most common method because it is difficult to select reference tissue. Tracers were evaluated to find the most suitable compartment model for precise quantification [83,84,85,86]. Taking [^11^C]verapamil as an example, the one-tissue compartment model (1-TCM) has been identified as the most optimal [86]. In a study involving rats, it was observed that the influx rate constant K1 underwent the most significant change after P-gp inhibition [86]. Similarly, in non-human primates, the 1-TCM and K1 models have been found to be the most suitable for quantifying transporter function during short scan durations [87]. In the case of human studies, 1-TCM has proven effective in measuring P-gp function using a short scan time without influences from retention of the tracer and radioactive metabolites [88].

However, although until now most P-gp-related PET research using kinetic modeling has been aimed at developing methods for brain scanning, some researchers have also tried to quantify P-gp function in the eyes and lungs [89,90]. Nevertheless, kinetic studies to measure P-gp function in the heart have not been found in the literature. Only semi-quantitative studies using the tracer [^11^C]Rhodamine-123 have been identified in the heart. Compared to other control groups, P-gp inhibitors increased [^11^C]Rhodamine-123 uptake in the heart tissue [91]. Further research is needed to evaluate the kinetics of P-gp tracers in the heart and to develop a kinetic model to quantify P-gp function.

### 4.2. PET Tracers for Measuring P-gp Function

Many PET tracers are radiolabeled substrates and synthesized to measure P-gp function in vivo (Table 3). Most of these are well-known P-gp substrates labeled with short half-life isotopes, such as [^11^C]verapamil, [^11^C]dLop, and [^18^F]Gefitinib. These tracers can be effectively transported out of cells by P-gp. Quantitative analysis can be used to indicate the efficiency of this efflux transport. Consequently, a PET tracer that acts as a substrate serves as an indicator of the P-gp function rather than its expression. Several studies have categorized substrate tracers as “avid” or “weak” based on their maximum transport capacity. The first-generation tracers designed to measure P-gp function are “avid”. However, the utilization of these tracers for evaluating P-gp function is hindered by their inherently low baseline uptake, which limits their ability to accurately assess P-gp overexpression. P-gp overexpression is commonly observed in pharmaco-resistance, as in epilepsy. To address this issue, weak substrate tracers have been specifically developed, with a higher brain uptake at baseline. These tracers aim to improve the evaluation of the P-gp function, as they are capable of measuring both increases and decreases in the P-gp function.

In general, a new P-gp tracer should preferably meet the following requirements [117,118]: (a) radiosynthesis should have a high radiochemical yield, and the tracer should be produced with high molar activity; (b) the tracer should have adequate specificity and sensitivity; (c) the tracer’s metabolism should be low during the scanning time; and (d) the tracer should have moderate plasma protein and blood cell binding. Below, we address the commonly used PET tracers for evaluating the function of P-gp.

#### 4.2.1. [^11^C]verapamil and (R)-[^11^C]verapamil

Verapamil itself acts both as an avid substrate and an inhibitor of P-gp, although it does not exhibit an inhibitory effect in the tracer doses utilized in PET. However, [^11^C]verapamil presents a challenge due to the differing pharmacokinetic properties of its enantiomers [92]. Some studies have also used racemic carbon-11-labeled verapamil ((R)-[^11^C]verapamil) in vivo to image P-gp expression. They discovered that, in comparison to (S)-verapamil, (R)-verapamil was less metabolized in humans [93,94]. Nowadays, (R)-[^11^C]verapamil is the most often used PET tracer to test P-gp function [118,119] in the brains of both animals [119,120,121] and humans [122,123,124,125].

However, in the heart, the uptake of [^11^C]verapamil was low and showed no differences in uptake between wild-type and mdr1a(−/−) mice [119]. Similarly, the uptake of both (R)- and (S)-[^11^C]verapamil in the heart showed no significant differences between wild-type mice and double knockout (dKO) mice [93]. A possible explanation for these results may be caused because of the low P-gp expression in the heart and the sensitivity of the tracer for metabolism. Therefore, [^11^C]verapamil may not be suitable for future cardiac P-gp investigation.

#### 4.2.2. [^18^F]MC225

[^18^F]MC225 (5-(1-(2-[^18^F] fluoroethoxy))-[3-(6,7-dimethoxy-3,4-dihydro-1H-isoquinolin-2-yl)-propyl]-5,6,7,8-tetrahydronaphthalene) has been developed and validated for quantitative assessment of P-gp function at the BBB. In 2021, Mossel et al. [22,126] conducted [^18^F]MC225 PET brain scans in a healthy human volunteer. Through blocking with P-gp inhibitor cyclosporin, the whole brain V_T_ changed from 4.38 at baseline to 5.48 [22]. These results demonstrated that [^18^F]MC225 is a promising tracer for assessing P-gp function in humans. According to an in vitro cardiovascular safety evaluation [127], the concentration of MC225 needed for P-gp function in vivo assessment does not cause cardiovascular effects.

Further, the expression of cardiac P-gp may up- or down-regulate with differences in diseases, drugs, age, and gender. [^18^F]MC225 is a weak substrate and can measure inducer effects on cardiac P-gp as well, something that may not be possible with avid subtract tracers [95,96]. Therefore, given its effectiveness in measuring the up and down-regulation of P-gp function, [^18^F]MC225 PET seems to be a promising tracer to study clinical assessment of cardiac P-gp function. [^18^F]MC225 may be promising for cardiac P-gp quantification. However, as there is no data available on cardiac P-gp measurements with [^18^F], more studies are needed.

#### 4.2.3. [^18^F] and [^11^C]gefitinib

Gefitinib is a tyrosine kinase domain inhibitor of the epidermal growth factor receptor (EGFR), which prevents signaling in target cancer cells with mutant and excessive EGFR. [^18^F] and [^11^C] gefitinib have been thought to be valuable instruments for evaluating the combined effects of P-gp and another high-capacity efflux transporter, BCRP at the BBB in vivo [112,113].

Gefitinib-labeled radiotracers may be promising for evaluating human cardiac P-gp and BCRP function. Kawamura et al. [113] identified that [^11^C]gefitinib could detect the cardiac P-gp and BCRP functions in the hearts of mice. They observed that the [^11^C]gefitinib uptake in the heart was increased from 1.39 to 2.26 after pretreatment with P-gp inhibitor cyclosporin [113]. However, no clinical trials have been performed on gefitinib-labeled radiotracers, and, currently, both the validity and safety of these tracers in humans are unclear. In the future, more research is needed for the verification of human cardiac transporter (P-gp and BCRP) function.

#### 4.2.4. [^18^F]MPPF

The tracer [^18^F]MPPF (fluorine-18-labeled 4-(2′-methoxyphenyl)-1-[2′-(N-2″-pirydynyl)-p-fluorobenzamido] ethylpiperazine) was designed to measure 5-hydroxytryptamine (1A) (5-HT1A) receptor with PET [128]. In 2000, Passchier et al. [97] confirmed in a preclinical PET study that [^18^F]MPPF was also a P-gp substrate, and the tracer was developed for evaluation of brain P-gp function [98,129,130].

In a preclinical study, Laćan et al. detected that heart uptake of [^18^F]MPPF was increased in the heart 0.7 fold at 2.5 min, and 0.6 fold at 5 min, after injection of P-gp inhibitor cyclosporin [98], indicating that cardio-P-gp function could be inhibited and the change could be detected by [^18^F]MPPF-PET.

However, the clinical translation of [^18^F]MPPF for human cardiac P-gp evaluation remains debatable. Although the use of [^18^F]MPPF for 5-HT1A receptor evaluation in humans was successful, indicating the safety [131], some studies show that MPPF is not a P-gp substrate in non-human primates, which may be due to the differences in species [132,133]. There is no literature demonstrating the use of this tracer in human P-gp or cardiac P-gp. Therefore, whether [^18^F]MPPF is suitable for cardiac P-gp remains unclear.

#### 4.2.5. [^11^C]rhodamine-123

Rhodamine-123 was initially developed as a vital fluorescent dye and was originally a useful probe for monitoring the abundance and activity of mitochondria. Multiple investigators have recently reported that it is also a substrate of P-gp, which has led to its use to screen cancer drugs that serve as substrates for P-gp [134]. Currently, rhodamine-123 has been radio-labeled with [^11^C] and [^131^I] as PET and SPECT tracers, respectively, for evaluating P-gp function.

In rodent studies, [^11^C]rhodamine-123 has demonstrated good performance as a PET tracer. Bao et al. [91] observed that P-gp inhibitor led to an increase in uptake of [^11^C]rhodamine-123 in heart and cardiac tissue to plasma ratios compared to other groups. Furthermore, in mouse myocardium, 97% of [^11^C]rhodamine-123 remained unchanged at 30 min after administration. These findings suggest that rhodamine-123 is a substrate for P-gp in rodents, which makes [^11^C]rhodamine-123 a promising tracer for evaluating cardiac P-gp function.

Further research is needed to determine the applicability of [^11^C]rhodamine-123 for assessing human cardiac P-gp function. [^11^C]rhodamine-123 is the first PET tracer to be systematically investigated. However, these studies are limited to preclinical experiments. No studies have investigated the sensitivity of [^11^C]rhodamine-123 to human P-gp. Additionally, the safety of [^11^C]rhodamine-123 in humans has not been identified, although rhodamine-123 was found to be safe at a dose lower than a 96 mg/m^2^ body surface area in a phase I clinical trial [135].

#### 4.2.6. [^11^C]loperamide and [^11^C]dLop

Loperamide is initially used to treat dysentery and acts on µ-opiate receptors in the intestine. Recently, [^11^C]loperamide was found to be an avid substrate for P-gp at the BBB [99]. N-[^11^C]desmethyl loperamide ([^11^C]dLop), its putative radioactive metabolite and also avid P-gp substrate, may be superior in measuring P-gp function, because the radio metabolites of ^11^C-dLop, especially from further demethylation, have minimal entry into the brain. Many preclinical and clinical [101] studies have since successfully evaluated the P-gp function in the BBB with [^11^C]dLop.

Unfortunately, hardly any data describe cardiac P-gp function with [^11^C]loperamide and [^11^C]dLop. The application of these tracers in cardiac P-gp function needs further research. Given the minimal metabolite presence, [^11^C]dLop may offer advantages in the cardiac context.

#### 4.2.7. [^11^C]metoclopramide

Metoclopramide is a dopamine receptor antagonist and is used to treat nausea and vomiting in patients with gastroesophageal reflux disease or diabetic gastroparesis. It has been found that [^11^C]metoclopramide acts as a weak substrate and is a useful PET tracer for evaluating BBB P-gp function. Tariquidar significantly increased the uptake of [^11^C]metoclopramide in the brains of rats, with a 4.4-fold increase in K1 and a 2.3-fold increase in binding potential (K3/K4) [103]. In non-human primates, tariquidar led to a 1.28-fold increase in K1 and a 1.64-fold decrease in K2 [104]. Findings show that [^11^C]metoclopramide is a useful tracer for assessing P-gp function at the BBB in humans. However, its utility in assessing cardiac P-gp function is limited, and more research in this area is required.

#### 4.2.8. [^18^F]AVT-011

[2-(4{2-[6-(2-[^18^F]fluoroethoxy)-7-methoxy-3,4-dihydro-1H-isoquinolin-2-yl]ethyl}phenylcarbamoyl)-4,5-dimethoxyphenyl]amide, also known as [^18^F]AVT-011, is a radiotracer that acts as a substrate for ABCB1(P-gp) and ABCG2(BCRP). Kannan et al. [114] evaluated it in preclinical models and found that the tracer was able to quantify transporter function at the BBB and in tumors that express P-gp. They also conducted a biodistribution evaluation, which revealed an increasing trend in the uptake of radioactivity in the heart, from 1.7 ± 0.2 in wild-type mice to 2.0 ± 0.1 (%ID/g·min) in ABCB1a/b^−/−^ mice, although this difference was not statistically significant. Still, the findings suggest that [^18^F]AVT-011 may be a promising PET tracer for future cardiac function detection, and its application in human P-gp deserves further investigation.

#### 4.2.9. [^11^C]tariquidar, [^11^C]elacridar, and [^11^C]laniquidar

Third-generation P-gp inhibitors, like tariquidar, are labeled with C-11 and developed as a PET tracer for measuring ABC transporters. In contrast to P-gp substrate radiotracers, [^11^C]tariquidar is intended to attach to P-gp without being transported by it, allowing mapping of P-gp expression or distribution. Bauer et al. [110] developed [^11^C]tariquidar and suggested that it interacts specifically with P-gp and BCRP in wild-type and Mdr1a/b(−/−) mice in the BBB. In another study, [^11^C]tariquidar successfully measured hepatic canalicular P-gp and BCRP transport function in humans and mice [111].

Elacridar, like tariquidar, is also a third-generation P-gp inhibitor. It has been radiolabeled as [^11^C]elacridar to access the P-gp function in BBB P-gp evaluation and performs well in rats and dKO mice models [108]. In clinical studies, both [^11^C]elacridar and [^11^C]tariquidar showed good biodistribution and safety in twelve healthy volunteers [109]. These results for [^11^C]tariquidar and [^11^C]elacridar show that both probes are promising for clinical applications. Research regarding whether these tracers are suitable for P-gp imaging in the heart needs to be performed in both animals and humans.

Laniquidar is also a third-generation MDR modulator that was labeled with C-11 and developed as a P-gp tracer. [^11^C]laniquidar was evaluated in a preclinical study [107]. There was no significant change in [^11^C]laniquidar uptake in cardiac tissues 30 min after intravenous administration in cyclosporin A and valspodar-pretreated rats compared to control rats, according to biodistribution data. Therefore, it seems not to be a suitable tracer for cardiac P-gp imaging.

#### 4.2.10. [^11^C]BMP

Unlike other tracers, 6-bromo-7-[^11^C]methylpurine ([^11^C]BMP) is a prodrug that enters tissue by passive diffusion and is transformed by glutathione-S-transferases into its glutathione conjugate S-(6-(7-[^11^C]methylpurinyl)) ([^11^C]MPG) glutathione, which is an ABCC1(MRP1) substrate. The efflux rate of [^11^C]MPG can thus reflect the MRP1 function in the organ. [^11^C]BMP has been used in the brain, kidneys, and lungs to access MRP1 function in mice models [136,137,138]. In a [^11^C]BMP-dynamic study [115], the efflux rate in Mrp1-gene knockout mice showed a 90% reduction compared to wild-type mice. Noteworthy, in another study [116], compared with wild-type rats, Abcc1(−/−) rats had 352% higher lung exposure (AUC_lung_) and 86% lower elimination slope (k_E,lung_) after intratracheal administration, showing that pulmonary-given [^11^C]BMP can also quantify MRP1 activity in the lungs of rats. Based on the present findings, [^11^C]BMP-PET appears suitable for exploring the influence of disease, genetic polymorphisms, or concurrent medication use on MRP1 function.

Unfortunately, there is a lack of data demonstrating the cardiac application of [^11^C]BMP-PET. Given the crucial role of MRP1 in safeguarding against anthracyclines, mitoxantrone, and other cardiotoxic drugs in cardiac function, further and more comprehensive research is essential to evaluate the efficacy of [^11^C]BMP-PET in assessing cardiac protective functions.

## 5. Discussion

### 5.1. Opportunities

Given that many cardiovascular drugs have an affinity for ABC transporter and act as substrates, evaluating cardiac P-gp function for medical drug treatment is clinically of interest. PET imaging could provide functional information about the ABC transporter in the human heart. This noninvasive imaging technique allows researchers to investigate the effects of altered ABC transporter function on drug distribution to the heart caused by DDIs, diseases, and genetic polymorphisms. Furthermore, by focusing on drug accumulation in cardiac tissue, this approach may allow for early prediction of the biodistribution of new drug candidates, thereby increasing their chances of success in development; it may also enable the identification of patients prone to development of cardiotoxicity due to (toxic) medical drugs used in oncology.

In our review, we have summarized P-gp tracers that were originally developed to assess cerebral P-gp function and further research is often necessary to translate these tracers into effective tools for clinical cardiac P-gp assessment. P-gp tracers, such as [^11^C]elacridar, [^11^C]tariquidar, [^11^C]verapamil, [^11^C]metoclopramide, [^11^C]dLop, and [^18^F]MC225 have already been extensively studied in a clinical setting and shown to be effective and sensitive for measuring P-gp function at the human BBB. However, their efficacy in evaluating cardiac P-gp function remains unknown. Other tracers, such as [^11^C]gefitinib, [^11^C]BMP, and [^11^C]rhodamine-123, have been identified as useful in assessing cardiac P-gp in animal models, but their sensitivity to human P-gp has yet to be investigated.

Currently, [^18^F]MC225 and [^11^C]rhodamine-123 appear to be promising tracers for future clinical use. [^11^C]rhodamine-123 was the first tracer to be systematically evaluated for cardiac P-gp, and has shown good performance in the heart. [^18^F]MC225 was found to be sensitive to measure changes in human P-gp at the BBB after inhibition with cyclosporin [22]. As a weak substrate, the capability to access up- and down-regulation of the P-gp function is another benefit of [^18^F]MC225, which is particularly useful for the clinical assessment of complex cardiac P-gp function. In addition, Fluorine-18 has a longer half-life, thereby providing higher imaging quality and easier clinical use of [^18^F]MC225 compared with carbon-11 labeled radiotracers, making the tracer promising for future use. The variability of P-gp among individuals suggests that [^18^F]MC225 and [^11^C]rhodamine-123 have the potential to accurately assess P-gp function in each individual heart. This capability could then guide more precise medication use, tailoring treatment strategies based on individual variations in cardiac P-gp function.

In addition to animal and clinical trials, there is growing interest in using in vitro models for combinatorial drug testing. Although preclinical animal studies can evaluate new PET tracers in vivo, they may not accurately reflect human heart physiology due to species differences. Engineered heart tissues (EHTs), 3D organ-on-chip models of the heart, have recently imitated human heart conditions by including cardiac myocytes and vascular endothelial cells produced from human pluripotent stem cells (hPSCs) [139]. Without a need for animal models, this model could make it possible to test cardiac P-gp function realistically. Therefore, these EHTs can be combined with radioactive tracers and employed to screen P-gp-related drugs and assess their cardiotoxicity.

### 5.2. Challenges

Currently, PET research on the function of ABC transporters is focused mainly on the BBB. Examples of tracer uptake in the heart are available mostly derived from biodistribution data. Tracers specifically designed to assess cardiac transporter function are limited. Re-evaluating BBB P-gp PET tracers in the heart could be an effective and useful approach. During the experimental setup, special attention must be given to the differences between the heart and the BBB.

Firstly, P-gp expression in the heart is much lower than in the BBB [17,18]. Consequently, the functional disparities in P-gp may not be as pronounced as observed in the BBB. As a result, a more precise quantification method is essential, favoring kinetic modeling analysis over semi-quantitative methods like SUVs. Secondly, compared to the stationary brain, the rapid beating of the heart poses challenges to imaging analysis and data collection. Reliable strategies to compensate for cardiac and respiratory motion can be used to minimize motion artifacts, such as ECG triggering to synchronize image acquisition with the cardiac cycle, and navigator-gated acquisitions to suppress respiratory motion [140]. Thirdly, unlike the brain, distinguishing the heart between the blood pool and the heart tissue is another challenging aspect of cardiac scans. (PET/)MRI or (PET/)CT could be more helpful in the location of cardiac tissue boundaries than normal PET. Fourth, regarding the formation of radioactive metabolites, unlike the brain, which is shielded by the BBB, the heart remains in the blood pool. The penetration of polar radioactive metabolites in the brain is low, and therefore has less influence on quantification of the PET data. In the case of quantification of P-gp in the heart, radioactive metabolites in blood could influence quantification. To tackle this issue, it is crucial to thoroughly examine how the tracer is metabolized in vivo. To improve the scanning process, use early-stage data and exact quantification approaches such as kinetic modeling and parametric imaging when metabolite levels are at their lowest.

In vitro assays and in vivo models can also pose challenges during the process of a study. In preclinical animals, it is vital to consider gene differences between animal and human models, especially when it comes to ABC transporters. ABCB1 encodes the human P-gp, whereas rodents have two drug-transporting P-gp homologs, Mdr1a and Mdr1b, which share around 85% and 87% amino acid identity, respectively. Rhesus monkeys (93%) and chimps (97%) are more similar to humans [141]. These gene differences could explain the disparities in animal and primate studies; this applies especially when some tracers work well in rodents but not primates, consequently exacerbating the difficulties associated with clinical translation.

Human heart on chips and other in vitro models offer the potential for screening P-gp-related drugs and assessing cardiotoxicity alongside radioactive tracers. However, several challenges impede their effective use. Firstly, the small size of the organ-on-chip tissue poses a limitation due to current imaging devices for measuring PET tracer uptake in small animals with limited spatial resolution [142]. Specialized detectors and software are essential for optimal results. Additionally, material properties present obstacles; the commonly used polydimethylsiloxane (PDMS) material in chips tends to be adhesive to many PET tracers, complicating tissue uptake quantification [143]. Moreover, the current heart-on-chip model differs somewhat from normal human heart tissue, and its relative immaturity compared to the adult myocardium restricts its applicability in cardiac research [143].

## 6. Conclusions

In this review, we evaluate and explore the PET imaging applications of ABC transporters in the heart. The use of PET imaging to measure ABC transporters in the heart advances our understanding of drug distribution, concentration, and toxicity, and thereby facilitates more precise medication use. This technique may also allow for early prediction of the biodistribution of potential new drug candidates, with a particular emphasis on their accumulation in cardiac tissue, leading to improved success rates in drug development. However, there is currently limited research in this area. Further investigation should prioritize the development of safe, effective, and cardiac-specific P-gp function PET tracers, which can be translated into clinical practice for the evaluation and assessment of cardiac P-gp function.

## Figures and Tables

**Table 2 pharmaceuticals-16-01715-t002:** P-gp substrates in different types of drugs.

Cardiac Agents	P-gp Substrate	Ref.
Antiarrhythmic agents	Bepridil, Digoxin, Quinidine, Verapamil	[12,48]
Anticoagulant agents	Apixaban, Dabigatran, Rivaroxaban, Edoxaban, Warfarin	[50]
Antihypertensive agents	Aliskiren, Ambrisentan, Amlodipine, Celiprolol, Debrisoquine, Diltiazem, Labetalol, Losartan, Nadolol, Propranolol, Talinolol, Timolol.	[12,62,63]
Antiplatelet agents	Clopidogrel, Ticagrelor	[64]
Cholesterol-lowering drugs	Atorvastatin, Lovastatin, Simvastatin	[65]

**Table 3 pharmaceuticals-16-01715-t003:** Existing tracers for measuring P-gp function in animal and clinical studies.

Chemical Structure	Biomarker	Targeted Structures	Radionuclide	Animal Experiments	Clinical Trials	Ref.
BBB	Heart
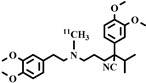	ABCB1 (P-GP)	substrate and inhibitor	Verapamil	^11^C	√ ^1^	×	√	[92]
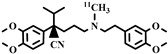	(R)-Verapamil	^11^C	√	×	√	[93,94]
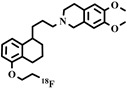	ABCB1 (P-GP)	substrate	MC225	^18^F	√		√	[22,95,96]
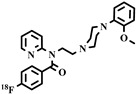	ABCB1 (P-GP)	substrate	MPPF	^18^F	√	√		[97,98]
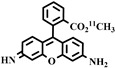	ABCB1 (P-GP)	substrate	Rhodamine-123	^11^C	×	√		[91]
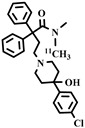	ABCB1 (P-GP)	substrate	Loperamide	^11^C	√		√	[99,100]
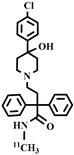	dLop	^11^C	√		√	[100,101,102]
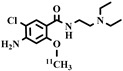	ABCB1 (P-GP)	substrate	Metoclopramide	^11^C	√		√	[103,104,105,106]
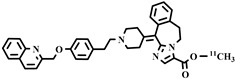	ABCB1 (P-GP)	inhibitor	Laniquidar	^11^C	√	×		[107]
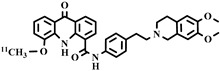	ABCB1 (P-GP)	inhibitor	Elacridar	^11^C	√		√	[108,109]
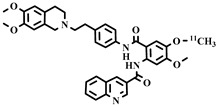	ABCG2 (BCRP)	inhibitor	Tariquidar	^11^C	√		√	[110,111]
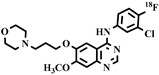	ABCB1 (P-GP)ABCG2 (BCRP)	substrate	Gefitinib	^18^F	√			[112]
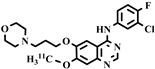	^11^C	√	√		[113]
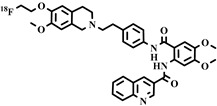	ABCB1 (P-GP)ABCG2 (BCRP)	Substrate	AVT-011	^18^F	√	×		[114]
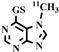	ABCC1(MRP1)	Substrate	BMP	^11^C	√			[115,116]

^1^ The symbol “√” signifies the use of tracers in those fields with promising results, while “×” indicates the use of tracers in those fields with insignificant results. A blank space indicates the absence of related research.

## Data Availability

Data sharing is not applicable.

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
