# Peer review of "Cardiac PET Imaging of ATP Binding Cassette (ABC) Transporters: Opportunities and Challenges"

_pharmaceuticals, 2023, doi:10.3390/ph16121715_

Round 1
Reviewer 1 Report
Comments and Suggestions for Authors
Dear Authors,
This review is a very well written update on clinical applications of radiotracers. Positive emission tomography is one of the most powerful, noninvasive technique to understand the physiological events. This review in the context of ABC transporters is very relevant. In human, 49 ABC transporters have been discovered. They transport a variety of compounds including proteins, lipids, and drugs and serve an important function in protecting organs such as the intestines, liver, brain, eyes, testicles, and heart against hazardous chemicals.
Recently, interest in ABC transporters and cardiology has been increasing. ABCC5 (multidrug resistance proteins 5, MRP5) and ABCG2 (breast cancer resistance protein, BRCP). They act as a functional barrier between the blood and other cells within the heart, such as cardiomyocytes and myofibroblasts, limiting medication and other xenobiotic entry into the heart.
P-gp is one of the most essential efflux transporters . Many commonly employed cardiovascular pharmaceuticals, such as β-blockers, cardiac glycosides, and Ca2+ channel blockers, are transported by P-gp and are known as P-gp substrates. P-gp transports these substrates, and any changes in P-gp function can impact the bioavailability of these pharmaceuticals.
To study the qualitative and quantitative analysis of cellular metabolites an internal radioactive probe helps to address this in noninvasive PET scanning.

Very minor editing is required. The table indents has to be rechecked
Author Response
Thank you for your suggestions. I have adjusted the table indents accordingly.
Reviewer 2 Report
Comments and Suggestions for Authors
The review article "Cardiac PET Imaging of ATP Binding Cassette (ABC) Transporters: Opportunities and Challenges " by Wanling Liu et al. is an detailed review of PET radiotracers for ABC transport measurment. The article may be of interest to a wide readership. The length of the manuscript and references are appropriate. I believe that the manuscript can be accepted into Pharmaceuticals after the following issues have been resolved:
1. Some typos should be corrected. ("pregnant xenobiotic receptor")
2. I suggest to clarify the following sentence: "...an 96 mg/m2 in a phase I clinical trial [101] ." -> "...an 96 mg/m2 body surface area in a phase I clinical trial [101] . "
3. The structures of dLop and Loperamide are confused (Table 3)
Some typos should be corrected.
Author Response
Reviewer 2:
Comments and Suggestions for Authors
1.Some typos should be corrected. ("pregnant xenobiotic receptor")
Thank you for the helpful comments. Corrections are made and changed it to “pregnane xenobiotic receptor” in line 243.
- I suggest to clarify the following sentence: "...an 96 mg/m2 in a phase I clinical trial [101] ." -> "...an 96 mg/m2 body surface area in a phase I clinical trial [101] . "
Thank you for your suggestions. The sentence is changed to “a 96 mg/m2 body surface area in a phase I clinical trial” in lines 473-474.
- The structures of dLop and Loperamide are confused (Table 3)
Thanks for the comments. The chemical structure is corrected see table 3.
Reviewer 3 Report
Comments and Suggestions for Authors
Abstract:
- “in vitro" and "in vivo" should be italicized to conform to scientific writing standards.
- Consider briefly mentioning the importance of ATP-binding cassette (ABC) transporters in cardiac drug delivery for more context in the abstract.
Introduction:
- In the introduction, consider providing a more explicit statement about the significance and motivation of this review to engage readers.
- Mention the abbreviation ABC along with its full form when it is first introduced.
- In the phrase "transport chemicals across cells and other membranes," you could specify that these are biological membranes.
- The sentence "Researchers have discovered their expression in the heart" could be more specific about which transporters are being referred to here.
- Use consistent terminology, such as ABCB1 and P-gp, throughout the text to avoid confusion.
- The phrase "and other xenobiotic entry into the heart" might be clearer as "and the entry of foreign substances into the heart."
- The section that discusses the importance of P-gp in cardiotoxicity could benefit from a more structured approach, introducing the topic, explaining its importance, and transitioning to the next point.
- Consider breaking down the lengthy sentence mentioning chemotherapy and monoclonal antibodies into two or more sentences for clarity.
- In the sentence, "due to the longer retention of the toxic agent in cardiomyocytes," it would be helpful to specify which "toxic agent" is being referred to.
ABC Transporters of the Heart:
- The introduction of ABC transporters in this section could be more comprehensive, briefly explaining their structure and how they function in the body.
- The mention of "Juliano et al." could be expanded to include more details about the study where P-gp was discovered.
- Table 1: Consider specifying the diseases in the table header for better clarity.
- Clarify what "atherosclerosis" and "coronary heart disease" are for readers who may not be familiar with these terms.
- The sentence "ABCC8 and ABCC9 are vital components of the ATP-sensitive potassium (KATP) channels in the heart" could benefit from explaining the role of these channels in the heart.
Regulation of P-gp in the Heart:
- Provide more context on the importance of P-gp regulation before discussing substrates, inhibitors, and inducers.
- Define "MDR" for readers unfamiliar with the term.
- Consider adding a sentence or two to introduce the term "modulators" as it may not be immediately clear what they are.
- Consider breaking down the section on P-gp inhibitors into a few sentences for better readability.
- When discussing P-gp substrates and inhibitors, include a brief description of what these drugs are used for, and their clinical significance.
- In the sentence, "The relationship between the substrate and inhibitor is complex and worthy of note," it would be helpful to briefly explain this complexity.
- The term "anticancer medication sensitivity" could be further explained for clarity.
- The section on "Inducers and activators of P-gp" could benefit from a clear explanation of what PXR and CAR are.
- Specify the implications of "enhanced P-gp expression" in "P-gp inducers enhance both its expression and function."
Heart diseases affecting P-gp function:
- Consider briefly explaining what "dilated non-ischemic cardiomyopathy" and "ischemic cardiomyopathy" are for clarity.
- In the sentence, "pathological conditions can alter P-gp expression and function," provide more specific examples of these conditions.
- Explain "hypoxia-inducible factor 1a (HIF-1α)" briefly, as it might not be immediately clear to all readers.
Other factors affecting P-gp function:
- Elaborate on the statement "Polymorphism in MDR-1 exon 26 (C3435T) may influence intestinal P-gp expression and function" to clarify what this means.
- Specify what the impact of "polymorphism in MDR-1 exon 26 (C3435T)" has on P-gp.
PET Imaging Section:
- Consider mentioning that "P-gp" stands for P-glycoprotein the first time you use the acronym to provide clarity.
- In the sentence "Biomathematical modeling is used to derive in vivo measures from radiotracer concentration-time curves," specify what "in vivo measures" refer to. Be more specific about what this modeling accomplishes.
- The phrase "making it a valuable tool for translational studies" could be clearer. Translational studies of what? It might be beneficial to clarify the applications of this technique further.
- Consider rephrasing the sentence "PET is also particularly useful and well-validated in non-invasive clinical investigation of vital organs such as the heart" for better clarity.
Methods to Measure ABC Transporters Function with PET Section:
- Clarify that "SUV" stands for Standardized Uptake Value when first mentioned.
- In the sentence "However, although until now most research using kinetic modeling has been aimed at developing methods for brain scanning," specify what specific research you are referring to.
- The sentence "Only semi-quantitative studies using SUV have been found in the heart with the tracer of [11C]Rhodamine-123" could be more explicit about what these studies have discovered.
PET Tracers for Measuring P-gp Function Section:
- The sentence "Most of these are well-known P-gp substrates labeled with short half-life isotopes, such as 11C or 18F" could be more informative by providing examples of these substrates.
- In the sentence "The most common used tracers are addressed below," it might be more precise to specify the most commonly used tracers for what purpose.
- Clarify the specific applications of [11C]verapamil and (R)-[11C]verapamil in P-gp assessment in the heart.
- In the section about [18F]MC225, clarify the significance of "cyclosporin" and how it relates to P-gp inhibition.
- Mention the context of BCRP in addition to P-gp when discussing [18F] and [11C]gefitinib.
- Provide more context about the [11C]rhodamine-123 tracer's potential applications in humans, especially in relation to cardiac P-gp function.
- Discuss the significance of [11C]loperamide and [11C]dLop as tracers for assessing cardiac P-gp.
- Clarify the relevance of [11C]BMP in measuring ABCC1 function in the heart and explain its potential importance.
Discussion Section:
- Consider clarifying the term "DDIs" the first time it's mentioned. Also, spell out "drug-drug interactions" for the sake of clarity.
- In the discussion of challenges, consider elaborating on "in vitro models" and their limitations and potential applications for assessing cardiac P-gp.
Extensive editing of English language required
Author Response
Reviewer 3:
Introduction:
- In the introduction, consider providing a more explicit statement about the significance and motivation of this review to engage readers.
Thank you for the suggestion. More explicit statement is made in the abstract in lines 79-84. “Therefore, employing PET imaging to assess cardiac P-gp will offer valuable insights for drug development and enhance the precision of medication usage. However, the current understanding in this area is limited. This review aims to explore the current applications of ABC transporter PET imaging, specifically focusing on P-gp in the heart. Additionally, we will discuss the opportunities and challenges in this evolving field.”
- Mention the abbreviation ABC along with its full form when it is first introduced.
Thanks for the suggestion. The full form of ABC transporters, “Adenosine triphosphate binding cassette (ABC),” is shown in lines 16 and 31.
.
- In the phrase "transport chemicals across cells and other membranes," you could specify that these are biological membranes.
Thanks, transport of chemicals across cells are specified and other membranes are mentioned as “biological membranes” in line 33.
- The sentence "Researchers have discovered their expression in the heart" could be more specific about which transporters are being referred to here.
Transporters are specified; “ABCB1 (P-glycoprotein, P-gp) [8, 9], ABCC5 (multidrug resistance proteins 5, MRP5) [10], and ABCG2 (breast cancer resistance protein, BRCP) [11].”in lines 92-95.
- Use consistent terminology, such as ABCB1 and P-gp, throughout the text to avoid confusion.
We used now consistent terminology; ‘P-gp’ in lines 40 and 46; “BCRP” in lines 42 and 106.
- The phrase "and other xenobiotic entry into the heart" might be clearer as "and the entry of foreign substances into the heart."
We use now "and the entry of foreign substances into the heart." In line 44-45,
- The section that discusses the importance of P-gp in cardiotoxicity could benefit from a more structured approach, introducing the topic, explaining its importance, and transitioning to the next point.
- Consider breaking down the lengthy sentence mentioning chemotherapy and monoclonal antibodies into two or more sentences for clarity.
We rewrote the discussion about cardiotoxicity in lines 50-55, as shown in the last question. "When P-gp is reduced, there is a potential for drug accumulation within the heart tissue. This elevated concentration of drugs in the cardiac tissue may cause decline in cardiac function, a phenomenon referred to as cardiotoxicity. "
- In the sentence, "due to the longer retention of the toxic agent in cardiomyocytes," it would be helpful to specify which "toxic agent" is being referred to.
We rewrote the discussion about cardiotoxicity in lines 53-54 and provided an example. "Example.g., Doxorubicin is a P-gp substrate that has been shown to accumulate in the hearts of mice lacking P-gp and to cause dose-dependent heart failure in cancer patients. "
ABC Transporters of the Heart:
- The introduction of ABC transporters in this section could be more comprehensive, briefly explaining their structure and how they function in the body.
Thanks for the suggestion. We added more details about ABC transporters in lines 100-103. "ABC transporters are membrane proteins that facilitate a variety of ATP-driven transport activities. They typically consist of a pair of nucleotide-binding domains (NBDs) and two transmembrane domains (TMDs). The TMDs are responsible for substrate specificity, while the NBDs exhibit a high degree of conservation across various ABC transporters. Positioned in the cytoplasm, the NBDs play a key role in transferring energy to facilitate the transport of the substrate across the membrane.”
- The mention of "Juliano et al." could be expanded to include more details about the study where P-gp was discovered.
Thanks for the suggestion. We added more details in lines 92-96.
"In 1976, Juliano et al. [25], through surface labeling studies, found a 170 K Daltons cell surface glycoprotein in the plasma membrane of ovary cells in colchicine-resistant Chinese hamsters. They designated it the P-glycoprotein (P-gp), which is the first ABC transporter. ".
- Table 1:Consider specifying the diseases in the table header for better clarity.
We specified the "atherosclerosis, " in lines 116-118. "Disorders in lipoprotein metabolism can contribute to the deposition of fatty substances in medium and large arteries, thereby causing atherosclerosis. This pathological condition, in turn, poses significant health risks.
- Clarify what "atherosclerosis" and "coronary heart disease" are for readers who may not be familiar with these terms.
We specified the "atherosclerosis, " in lines 116-118. "
We specified the "coronary heart disease" in lines 118-120. "Coronary heart disease (CHD) occurs when blood flow in the coronary arteries is reduced or blocked as a result of obstructive atherosclerosis. "
- The sentence "ABCC8 and ABCC9 are vital components of the ATP-sensitive potassium (KATP)channels in the heart" could benefit from explaining the role of these channels in the heart.
Thanks, we explained the role of these channels in lines 121-128.
"ABCC5 serves as an efflux pump for cyclic nucleotides, particularly 3',5'-Cyclic guanosine monophosphate (cGMP). This molecule holds significant importance as one of the primary second messengers pivotal in the regulation of cardiac contractility and electrophysiology.
ABCC8 and ABCC9 are vital components of the ATP-sensitive potassium (KATP) channels in the heart [39-41]. These KATP channels serve as unique cellular energy sensors, safeguarding cardiomyocytes, particularly during conditions such as ischemia [42]. "
Regulation of P-gp in the Heart:
- Provide more context on the importance of P-gp regulation before discussing substrates, inhibitors, and inducers.
We inserted the following lines " Up and down regulation P-gp activity within the body can significantly impact drug bioavailability, renal clearance, and peripheral tissue distribution. Therefore, it is crucial to understand how P-gp undergoes changes and is regulated by different influencing factors. " In lines 159-162.
- Define "MDR" for readers unfamiliar with the term.
We defined the MDR when we first mention it in lines 208-211."Extensive research and development efforts have been dedicated to inhibitors in the field of oncology. This is because of the association of P-gp with multi-drug resistance (MDR), a form of acquired resistance observed in cancer cells in response to chemotherapeutic treatments [66]. "
Consider adding a sentence or two to introduce the term "modulators" as it may not be immediately clear what they are.
We introduced modulators in lines 203-204 with "The presence of inhibitors, inducers, and/or activators, together referred to as modulators, affects P-gp expression and function levels."
- Consider breaking down the section on P-gp inhibitors into a few sentences for better readability.
We clarified the sentences about P-gp inhibitors in lines 212-217.
"The first generation includes verapamil, quinidine, and cyclosporin A. The second generation includes dexverapamil and dexniguldipine, which are more selective and have fewer adverse effects. Third-generation molecules, like Tariquidar, exhibit a strong affinity for P-gp at nanomolar doses. Fourth-generation compounds, including flavonoids, alkaloids, and terpenoids, have been produced to perform better in terms of efficacy and toxicity. "
- When discussing P-gp substrates and inhibitors, include a brief description of what these drugs are used for, and their clinical significance.
A description of these drugs and their clinical relevance is given in Lines 221-223 "verapamil is known as a calcium channel blocker, and is used in the cardiovascular diseases, including angina pectoris and hypertension. "
Lines 176-179 "Digoxin is used as a cardiac glycoside to treat heart failure and cardiac arrhythmias. Due to its the restricted therapeutic window, digoxin may lead to serious adverse drug reactions ranging from arrhythmia recurrence, failure of device-based treatment, cardiac failure, and death. "
- In the sentence, "The relationship between the substrate and inhibitor is complex and worthy of note," it would be helpful to briefly explain this complexity.
We explained them in lines 237-240 with "Several drugs, such as quinidine, verapamil, warfarin, and atorvastatin, are also known as inhibitors. The effect of these inhibitors is dose dependent. For example, tariquidar in lower concentrations inhibits P-gp while acting as a BCRP substrate. However, when its concentration is higher than 100 nM it inhibits both P-gp and BCRP. "
- The term "anticancer medication sensitivity" could be further explained for clarity.
We give a further explanation in lines 228-232 " The inhibitor nilotinib can effectively restrain the P-gp-mediated efflux of anticancer drugs. This inhibition leads to an increased concentration of anticancer drugs within the tumor, thereby enhancing the sensitivity of anticancer medication. "
- The section on "Inducers and activators of P-gp" could benefit from a clear explanation of what PXR and CAR are.
We explained this in lines 242-246. " The inducers are facilitated through the pregnane xenobiotic receptor (PXR) and the constitutive androstane receptor (CAR). Both PXR and CAR are nuclear receptors. Fol-lowing activation by P-gp inducers, they bind to the transcriptional binding sites of P-gp, thereby promoting the increased expression. "
- Specify the implications of "enhanced P-gp expression" in "P-gp inducers enhance both its expression and function."
We explained in lines 244-246. "Both PXR and CAR are nuclear receptors. Following activation by P-gp inducers, they bind to the transcriptional binding sites of P-gp, thereby promoting the increased expression. "
Heart diseases affecting P-gp function:
- Consider briefly explaining what "dilated non-ischemic cardiomyopathy" and "ischemic cardiomyopathy" are for clarity.
Explained in lines 257-261. “Lower expression levels of P-gp exhibited in nonischemic dilated cardiomyopathy, characterized by impaired heart systolic function without interruption of perfusion [73, 74]. This differs from ischemic cardiomyopathy, where systolic left ventricular dysfunction results from the interruption of blood perfusion.”
- In the sentence, "pathological conditions can alter P-gp expression and function," provide more specific examples of these conditions.
In lines 255, we changed the “pathological conditions” to diseases. The specific examples are "dilated non-ischemic cardiomyopathy" and "ischemic cardiomyopathy," which are explained after the sentences in lines 259-261.
- Explain "hypoxia-inducible factor 1a(HIF-1α)" briefly, as it might not be immediately clear to all readers.
We explained hypoxia-inducible factor 1a (HIF-1α) in lines 262-267.
"In a recent study by Auzmendi et al. [8], it was discovered that seizures induce ischemia–reperfusion injury, and this phenomenon is associated with an up-regulation of P-gp in cardiomyocytes. The observed increase in P-gp expression can be ascribed to the activation of hypoxia-inducible factor 1α (HIF-1α). Serving as a crucial transcriptional regulator, HIF-1α plays a central role in cellular adaptation to hypoxic conditions by overseeing the transcriptional activation of various genes, including ABCA1. "
Other factors affecting P-gp function:
- Elaborate on the statement "Polymorphism in MDR-1 exon 26 (C3435T)may influence intestinal P-gp expression and function" to clarify what this means.
We explained this part in lines 278-282. "Burk et al. [78] describe the discovery and distribution of 15 polymorphisms in the human ABCB1 gene that codes for P-gp, one of which, exon 26 (C3435T), is associated with P-gp expression levels and function in vivo. The C3435T mutation frequency is highly impacted by ethnicity, with African groups having higher frequencies than Caucasian and Asian populations [79]. "
Specify what the impact of "polymorphism in MDR-1 exon 26 (C3435T)" has on P-gp.
We explained the impact in lines 278-282. "Burk et al. [78] describe the discovery and distribution of 15 polymorphisms in the human ABCB1 gene that codes for P-gp, one of which, exon 26 (C3435T), is associated with P-gp expression levels and function in vivo. The C3435T mutation frequency is highly impacted by ethnicity, with African groups having higher frequencies than Caucasian/Asian populations [79]. "
PET Imaging Section:
- Consider mentioning that "P-gp" stands for P-glycoprotein the first time you use the acronym to provide clarity.
We mentioned "P-glycoprotein: in line 38.
- In the sentence "Biomathematical modeling is used to derive in vivo measures from radiotracer concentration-time curves," specify what "in vivo measures" refer to. Be more specific about what this modeling accomplishes.
Thanks. We explained "in vivo measures" in line 293-294: "PET stands as a valuable imaging technique for the in vivo assessment and quantification of P-gp function in both animals and humans. "
For the modeling we specific in lines 293-301. " PET stands as a valuable imaging technique for the in vivo assessment and quantification of P-gp function in both animals and humans. Kinetic modeling is used to derive in vivo measures from radiotracer concentration-time curves in plasma and tissue. Micro parameters like K1, k2 and the volume of distribution (VT), which reflects the P-gp function, can be calculated. PET is a non-invasive imaging technique and finds extensive application in both preclinical and clinical research, making it a valuable tool for translational studies focused on P-gp substrates and regulatory drugs. PET proves to be highly beneficial and is well-validated for non-invasive clinical investigations of vital organs such as the heart. "
- The phrase "making it a valuable tool for translational studies" could be clearer. Translational studies of what? It might be beneficial to clarify the applications of this technique further.
We clarified the applications in lines 293-301. " Kinetic modeling is used to derive in vivo measures from radiotracer concentration-time curves in plasma and tissue. Micro parameters like K1, k2 and the volume of distribution (VT), which reflects the P-gp function, can be calculated. PET is a non-invasive imaging technique and finds extensive application in both preclinical and clinical research, making it a valuable tool for translational studies focused on P-gp substrates and regulatory drugs. PET proves to be highly beneficial and is well-validated for non-invasive clinical investigations of vital organs such as the heart. "
- Consider rephrasing the sentence "PET is also particularly useful and well-validated in non-invasive clinical investigation of vital organs such as the heart" for better clarity.
We rephrase it to "PET proves to be highly beneficial and is well-validated for non-invasive clinical investigations of vital organs such as the heart." In lines 300-301.
Methods to Measure ABC Transporters Function with PET Section:
- Clarify that "SUV" stands for Standardized Uptake Value when first mentioned.
We inserted Standardized Uptake Value in line 304.
- In the sentence "However, although until now most research using kinetic modeling has been aimed at developing methods for brain scanning," specify what specific research you are referring to.
Thanks. we added "P-gp-related" in line 323-324 with "However, although until now most P-gp-related PET research using kinetic modeling has been aimed at developing methods for brain scanning, some researchers have also tried to quantify P-gp function in the eyes and lung. "
- The sentence "Only semi-quantitative studies using SUV have been found in the heart with the tracer of [11C]Rhodamine-123" could be more explicit about what these studies have discovered.
We added the context of these studies in lines 326-329. "Only semi-quantitative studies using the tracer [11C]Rhodamine-123 have been identified in the heart [70]. Compared to other control group, P-gp inhibitors increased [11C]Rhodamine-123 uptake in the heart tissue. "
PET Tracers for Measuring P-gp Function Section:
- The sentence "Most of these are well-known P-gp substrates labeled with short half-life isotopes, such as 11C or 18F" could be more informative by providing examples of these substrates.
We changed the sentence to "Most of these are well-known P-gp substrates labeled with short half-life isotopes, such as [11C]verapamil, [11C]dLop and [18F]Gefitinib. " In line 333.
- In the sentence "The most common used tracers are addressed below," it might be more precise to specify the most commonly used tracers for what purpose.
We specified the purpose "below, we address the commonly used PET tracers for evaluating the function of P-gp. " In lines 359-360.
- Clarify the specific applications of [11C]verapamil and (R)[11C]verapamil in P-gp assessment in the heart.
Thank you for your insightful suggestions. Clarifying the specific applications of these tracers is highly valuable. Regrettably, the available literature in this field is quite limited. Recently, we have only found in vivo data about the uptake of [11C]verapamil in the heart, without specific applications for this tracer in cardiac contexts.
- In the section about [18F]MC225, clarify the significance of "cyclosporin" and how it relates to P-gp inhibition.
We added that Cyclosporin is a P-gp inhibitor in line 397.
- Mention the context of BCRP in addition to P-gp when discussing [18F] and [11C]gefitinib.
We explained that BCRP is another high-capacity efflux transporter in line 414.
- Provide more context about the [11C]rhodamine-123 tracer's potential applications in humans, especially in relation to cardiac P-gp function.
We added tracer's potential applications in humans in lines 595-599 "The variability of P-gp among individuals suggests that [18F]MC225 and [11C]rhodamine-123 have the potential to accurately assess P-gp function in each individual. This capability could then guide more precise medication use, tailoring treatment strategies based on individual variations in P-gp function. "
- Discuss the significance of [11C]loperamide and [11C]dLop as tracers for assessing cardiac P-gp.
We discuss and inserted in line 485-487 the following: "The application of these tracers in cardiac P-gp function needs further research. Given the minimal metabolite presence, [11C]dLop may offer advantages in the cardiac context. "
- Clarify the relevance of [11C]BMP in measuring ABCC1 function in the heart and explain its potential importance.
We inserted the relevance in lines 553-557. " Unfortunately, there is a lack of data demonstrating the cardiac application of [11C]BMP-PET. Given the crucial role of MRP1 in safeguarding against anthracyclines, mitoxantrone, and other cardiotoxic drugs in cardiac function, further and more com-prehensive research is essential to evaluate the efficacy of [11C]BMP-PET in assessing cardiac protective functions. "
Discussion Section:
- Consider clarifying the term "DDIs" the first time it's mentioned. Also, spell out "drug-drug interactions" for the sake of clarity.
We clarified the term DDI in 3.1 in lines 218-210. "In clinical practice, first-generation P-gp inhibitors play a crucial role in drug-drug interactions (DDI). This implies that the effects of those drugs can modify the impact of others, thereby influencing their respective therapeutic domains. "
- In the discussion of challenges, consider elaborating on "in vitro models" and their limitations and potential applications for assessing cardiac P-gp.
We added the limitations in lines 648-658. " Human heart-on-chips and other in vitro models offer potential for screening P-gp-related drugs and assessing cardiotoxicity alongside radioactive tracers. However, several challenges impede their effective use. Firstly, the small size of the organ-on-chip tissue poses a limitation due to current imaging devices for measuring PET tracer uptake in small animals with limited spatial resolution [145]. Specialized detectors and software are essential for optimal results. Additionally, material properties present obstacles; the commonly used polydimethylsiloxane (PDMS) material in chips tends to be adhesive to many PET tracers, complicating tissue uptake quantification [146]. Moreover, the current heart-on-chip model differs somewhat from normal human heart tissue, and its relative immaturity compared to the adult myocardium restricts its applicability in cardiac research [146]. "
And the potential applications in line 600-608 " In addition to animal and clinical trials, there is growing interest in using in vitro models for combinatorial drug testing. Although preclinical animal studies can evaluate new PET tracers in vivo, they may not accurately reflect human heart physiology due to species differences. Engineered heart tissues (EHTs), 3D organ-on-chip models of the heart, have recently imitated human heart conditions by including cardiac myocytes and vascular endothelial cells produced from human pluripotent stem cells (hPSCs) [142]. Without a need for animal models, this model could make it possible to test cardiac P-gp function realistically. Therefore, these EHTs can be combined with radioactive tracers and employed to screen P-gp-related drugs and assess their cardiotoxicity."
Reviewer 4 Report
Comments and Suggestions for Authors
The manuscript explores the ATP-binding cassette (ABC) transporters, like P-glycoprotein (P-gp), impact cardiac drug concentrations and cardiotoxicity. This review examines PET imaging's role in studying ABC transporters in the heart, emphasizing P-gp, along with discussing related opportunities and challenges. However, few comments are highlighted below for authors considerations
Comments for authors:
Title:
The title accurately reflects the content of the study and is concise and informative.
Abstract:
1. Even with the restrictions on the abstract length, It's crucial to specify the objectives of the review more explicitly. What are the research questions or areas of interest that will be explored? This would provide readers with a roadmap for what to expect
Introduction:
1. Provide concise definitions for specialized terms, such as "cardiotoxicity," especially if the target audience includes individuals who may not be well-versed in this field.
2. This section maintains clarity in conveying essential information about ABC transporters and their roles in cardiac tissues. It's well-structured and employs appropriate scientific terminology. However, the use of specific terms like "cGMP" and "KATP channels" may require brief explanations or references for readers who are not familiar with these terms.
3. Provide a brief explanation of some specialized terms (e.g., "cGMP" and "KATP channels") for readers who may not be well-versed in this field.
4. Specify the scope and objectives of the review earlier in the introduction. Clearly state what specific questions or areas the review will address.
Section 3:
5. Provide a brief explanation of some specialized terms (e.g., "cGMP" and "KATP channels") for readers who may not be well-versed in this field.
6. Specify the scope and objectives of the review earlier in the introduction. Clearly state what specific questions or areas the review will address.
Section 4:
7. Provide a brief explanation of some specialized terms (e.g., "cGMP" and "KATP channels") for readers who may not be well-versed in this field.
8. Specify the scope and objectives of the review earlier in the introduction. Clearly state what specific questions or areas the review will address.
9. While the section provides an overview of various tracers and their applications, a more direct comparison between different tracers in terms of sensitivity, specificity, and clinical feasibility would enhance its utility for researchers and clinicians. This would assist readers in making informed choices when selecting a tracer for specific applications
10. The review of PET tracers is comprehensive, providing examples and insights into their effectiveness in assessing P-gp function. However, there is room for a more critical analysis of the limitations and challenges associated with these tracers. For example, the suitability of certain tracers like [11C]verapamil for cardiac P-gp assessment is questioned, but a more in-depth discussion of why this is the case would be beneficial.
Discussion:
11. The section is generally well-detailed, but it could benefit from specific examples or case studies to illustrate the points made. For instance, discussing a real-world scenario where the use of [18F]MC225 or [11C]rhodamine-123 was particularly informative for cardiac P-gp assessment would add depth to the discussion.
12. While the challenges section effectively addresses important considerations, it could be expanded to provide more in-depth solutions or potential research directions. For instance, discussing strategies for dealing with the lower P-gp expression in the heart or the issue of radioactive metabolites in blood influencing quantification could offer valuable insights.
Conclusion:
13. The discussion should conclude by summarizing the key findings and their implications, reiterating the importance of assessing cardiac P-gp function, and suggesting potential avenues for future research. A strong conclusion enhances the overall impact of the discussion section.
References:
14. The manuscript would benefit from a more comprehensive and up-to-date list of references. Ensure that all claims and findings are properly supported by references to the relevant literature. Additionally, ensure that the citation format is consistent throughout the manuscript.
Author Response
Review 4:
Comments for authors:
Abstract:
- Even with the restrictions on the abstract length, it’s crucial to specify the objectives of the review more explicitly. What are the research questions or areas of interest that will be explored? This would provide readers with a roadmap for what to expect.
Thank you for the suggestions; "we have incorporated additional sentences outlining the objectives in lines 22-26.” Using PET imaging to evaluate cardiac P-gp function provides new insights for drug development and improve the precise use of medications. Nevertheless, information in this field is limited. In this review, we aim to examine the current applications of ABC transporter PET imaging and its tracers in the heart, with a specific emphasis on P-gp. Furthermore, the opportunities and challenges in this novel field will be discussed.”
Introduction:
- Provide concise definitions for specialized terms, such as "cardiotoxicity," especially if the target audience includes individuals who may not be well-versed in this field.
We explained the " cardiotoxicity " in line 54-59. " When P-gp function is reduced, there is a potential for (toxic) drug accumulation within the heart tissue. This concentration of drugs in the cardiac tissue may cause decline in cardiac function, a phenomenon referred to as cardiotoxicity. For example, Doxorubicin is a P-gp substrate that has been shown to accumulate in the hearts of mice lacking P-gp and to cause dose-dependent heart failure in cancer patients."
We explained "NBDs and TMDs " in lines 88-92. " They typically consist of a pair of nucleotide-binding domains (NBDs) and two trans-membrane domains (TMDs). The TMDs are responsible for substrate specificity, while the NBDs exhibit a high degree of conservation across various ABC transporters. Positioned in the cytoplasm, the NBDs play a key role in transferring energy to facilitate the transport of the substrate across the membrane."
We explained the "atherosclerosis " and " coronary heart disease (CHD) " in lines 117-121. " Disorders in lipoprotein metabolism can contribute to the deposition of fatty substances in medium and large arteries, thereby causing atherosclerosis and potential vascular obstruct-ing resulting in brain or cardiac infarction. [37]. Coronary heart disease (CHD) negatively influences the heart function when blood flow in the coronary arteries is reduced or blocked as a result of obstructive atherosclerosis [38]. "
- This section maintains clarity in conveying essential information about ABC transporters and their roles in cardiac tissues. It's well-structured and employs appropriate scientific terminology. However, the use of specific terms like "cGMP" and "KATP channels" may require brief explanations or references for readers who are not familiar with these terms.
We explained the "cGMP and KATP" in lines 122-129. " ABCC5 serves as an export pump for cyclic nucleotides, particularly 3',5'-Cyclic guanosine monophosphate (cGMP). This molecule holds significant importance as one of the primary second messengers pivotal in the regulation of cardiac contractility and electrophysiology [26]. ABCC5 is found in cardiomyocytes as well as endothelial cells, and its expression is enhanced in ischemic cardiomyopathy, which is defined as systolic left ventricular failure in the presence of obstructive coronary artery disease. As a result, ABCC5-mediated cellular export might be a unique, disease-dependent mechanism for cGMP removal from cardiac cells.
ABCC8 and ABCC9 are vital components of the ATP-sensitive potassium (KATP) channels in the heart [39-41]. These KATP channels serve as unique cellular energy sensors, safeguarding cardiomyocytes, particularly during conditions such as ischemia [42]. "
- Provide a brief explanation of some specialized terms (e.g., "cGMP" and "KATP channels") for readers who may not be well-versed in this field.
We explained the terms "cGMP and KATP" in lines 122-129. " ABCC5 serves as an export pump for cyclic nucleotides, particularly 3',5'-Cyclic guanosine monophosphate (cGMP). This molecule holds significant importance as one of the primary second messengers pivotal in the regulation of cardiac contractility and electrophysiology [26]. ABCC5 is found in cardiomyocytes as well as endothelial cells, and its expression is enhanced in ischemic cardiomyopathy, which is defined as systolic left ventricular failure in the presence of obstructive coronary artery disease. As a result, ABCC5-mediated cellular export might be a unique, disease-dependent mechanism for cGMP removal from cardiac cells.
ABCC8 and ABCC9 are vital components of the ATP-sensitive potassium (KATP) channels in the heart [39-41]. These KATP channels serve as unique cellular energy sensors, safeguarding cardiomyocytes, particularly during conditions such as ischemia [42]. "
- Specify the scope and objectives of the review earlier in the introduction. Clearly state what specific questions or areas the review will address.
We specified the scope and objectives in lines 79-84 with "Therefore, employing PET imaging to assess cardiac P-gp will offer valuable insights for drug development and enhance the precision of medication usage. However, the current understanding in this area is limited. This review aims to explore the current applications of ABC transporter PET imaging, specifically focusing on P-gp in the heart. Additionally, we will discuss the opportunities and challenges in this evolving field. "
While the section provides an overview of various tracers and their applications, a more direct comparison between different tracers in terms of sensitivity, specificity, and clinical feasibility would enhance its utility for researchers and clinicians. This would assist readers in making informed choices when selecting a tracer for specific applications.
It would be valuable to conduct a comprehensive comparison of the sensitivity, specificity, and clinical feasibility of all the tracers. However, performing such a comparison is challenging due to the diverse testing models and analytical tools employed for each tracer. Currently, the available data only provides a head-to-head comparison of (R)-[11C]verapamil and [18F]MC225 in non-human primates, specifically focusing on the brain. Unfortunately, there is a lack of data on these tracers in cardiac applications. As a result, the limited availability of data hinders our ability to adequately identify and compare their sensitivity and specificity in these. Context.
a general summary, designating [18F]MC225 or [11C]Rhodamine-123 as the most promising tracers in lines 588-589. with "Currently, [18F]MC225 and [11C]rhodamine-123 appear to be promising tracers for future clinical use. "
The review of PET tracers is comprehensive, providing examples and insights into their effectiveness in assessing P-gp function. However, there is room for a more critical analysis of the limitations and challenges associated with these tracers. For example, the suitability of certain tracers like [11C]verapamil for cardiac P-gp assessment is questioned, but a more in-depth discussion of why this is the case would be beneficial.
We explained this issue in lines 374-378 with " A possible explanation for these results may be caused because the low P-gp expression in the heart and the sensitivity of the tracer for metabolism. Therefore, [11C]verapamil may not be suitable for future cardiac P-gp investigation. "
Discussion:
- The section is generally well-detailed, but it could benefit from specific examples or case studies to illustrate the points made. For instance, discussing a real-world scenario where the use of [18F]MC225 or [11C]Rhodamine-123 was particularly informative for cardiac P-gp assessment would add depth to the discussion.
We illustrate this item in lines 596-600 with “The variability of P-gp among individuals suggests that [18F]MC225 and [11C]rhodamine-123 have the potential to accurately assess P-gp function in each individual heart. This capability could then guide more precise medication use, tailoring treatment strategies based on individual variations in cardiac P-gp function.”
- While the challenges section effectively addresses important considerations, it could be expanded to provide more in-depth solutions or potential research directions. For instance, discussing strategies for dealing with the lower P-gp expression in the heart or the issue of radioactive metabolites in blood influencing quantification could offer valuable insights.
We added in strategies part about "lower P-gp expression" in lines 617-620 with "Firstly, P-gp expression in the heart is much lower than in the BBB [17, 18]. Consequently, the functional disparities in P-gp may not be as pronounced as observed in the BBB. As a result, a more precise quantification method is essential, favoring kinetic modeling analysis over semi-quantitative methods like SUV. "
We added strategies about "heart beating " in lines 620-625. "Secondly, compared to the stationary brain, the rapid beating of the heart poses challenges to imaging analysis and data collection. Reliable strategies to compensate for cardiac and respiratory motion can be used to minimize motion artifacts, such as ECG triggering to synchronize image acquisition with the cardiac cycle, and navigator-gated acquisitions to suppress respiratory motion [143]. "
We added strategies about "distinguishing heart tissue " in lines 629-630. "Thirdly, unlike the brain, distinguishing in the heart between the blood pool and the heart tissue is another challenging aspect of cardiac scans. (PET/) MRI or (PET/)CT could be more helpful in the location of cardiac tissue boundaries than normal PET. "
We added strategies "about radioactive metabolites" in lines 636-639. "To tackle this issue, it's crucial to thoroughly examine how the tracer is metabolized in vivo. To improve the scanning process, use early-stage data and exact quantification approaches such as kinetic modeling and parametric imaging when metabolite levels are at their lowest. "
Conclusion:
- The discussion should conclude by summarizing the key findings and their implications, reiterating the importance of assessing cardiac P-gp function, and suggesting potential avenues for future research. A strong conclusion enhances the overall impact of the discussion section.
We concluded in lines 573-574 with "In this review, we evaluate and explore PET imaging applications of ABC transporters in the heart. The use of PET imaging to measure ABC transporters in the heart advance our understanding of drug distribution, concentration, and toxicity, and thereby facilitate more precise medication use. "
References:
- The manuscript would benefit from a more comprehensive and up-to-date list of references. Ensure that all claims and findings are properly supported by references to the relevant literature. Additionally, ensure that the citation format is consistent throughout the manuscript.
Thanks, we add many up-to-date references and cite the references according to the Journal guide.
Round 2
Reviewer 3 Report
Comments and Suggestions for Authors
Accepted for publication.
Author Response
The report form does not contain any comments. Thank you for your agreement.
Reviewer 4 Report
Comments and Suggestions for Authors
The authors are commended for the revisions. No further comments!
Author Response

(The authors gave the same response as above.)
